# Anaemia in Lambs and Kids Reared Indoors on Maternal Milk and the Impact of Iron Supplementation on Haemoglobin Levels and Growth Rates

**DOI:** 10.3390/ani12141863

**Published:** 2022-07-21

**Authors:** James Patrick Crilly, Peter Plate

**Affiliations:** 1Royal Veterinary College, Hawkshead Lane, Hatfield, Hertfordshire AL9 7TA, UK; 2Royal Veterinary College, Regional Veterinary Centre South of England, Stinsford Business Centre, Kingston Maurward College, Dorchester DT2 8PY, UK; pplate@rvc.ac.uk

**Keywords:** iron, anaemia, goat kids, lambs, haemoglobin, growth rates

## Abstract

**Simple Summary:**

The study assessed anaemia (low haemoglobin levels) due to iron deficiency in new-born lambs and goat kids. Blood samples were taken from lambs and kids under different management systems at one month of age, and those reared indoors and suckling their mothers showed lower haemoglobin levels than those on milk replacer (which is fortified with iron) or those reared outdoors, indicating iron deficiency anaemia. As a follow-up, lambs and kids from those “at-risk” settings were enrolled on a trial injecting iron into half the animals in the first one to eight days of life. Those injected animals showed higher haemoglobin levels at one month of age (i.e., not getting anaemia) than untreated ones, and there was also a trend towards higher growth rates in treated animals, especially in twin lambs.

**Abstract:**

This study aimed to assess iron deficiency anaemia in new-born lambs and goat kids and was carried out in two parts: (1) Twenty blood samples were taken from one-month-old lambs and kids under different systems and were tested for haemoglobin. Three groups of lambs were compared: indoor reared on maternal milk, indoor reared on milk replacer, and outdoor reared on maternal milk. Indoor-reared kids were compared: those fed on maternal milk and fed on milk replacer. Indoor-reared kids and lambs on maternal milk showed significantly lower haemoglobin levels than those on milk replacer or reared outdoors. (2) On farms with indoor-reared lambs or goat kids on maternal milk, an intervention trial was carried out: animals were randomly assigned at 1–8 days of age to either receive 300 mg (lambs) or 150 mg iron (goat kids) as intramuscular iron dextran, and growth rates were compared after one and two months. Haemoglobin levels at one month were also compared in randomly selected animals from both groups. Treated lambs and kids showed higher haemoglobin levels at one month of age and a numerically increased growth rate that was statistically significant for twin lambs. Iron dextran improves haemoglobin levels in these animals and may lead to higher growth rates, especially in twin lambs.

## 1. Introduction

Iron has a multitude of functions in the body. Apart from its well-known role in oxygen transport as part of haemoglobin, it has also been shown to affect immune function [1] and—as part of anti-oxidant enzymes—red blood cell survival time in calves [2] and also intestinal development [3], brain development [3], and spatial cognition [4] of piglets. It is increasingly recognised that in a similar way as piglets [5], housed, whole-milk-fed calves are often iron deficient, with subsequent anaemia, poorer growth rates, and greater susceptibility to disease [6,7,8,9,10]. Consequently, calf milk replacers are generally supplemented with iron with a generally recommended inclusion rate of 100 mg/kg dry matter [11]. It is not known if the same trend widely seen in pigs and cattle is also evident in small ruminants, although anaemia in indoor-reared suckled lambs has been reported previously [12,13,14]. In goat kids, there is even less previous work, but Skrzypczak et al. [15] reported that plasma iron concentrations declined between 5 and 14 days of age, before rising again, in Polish Improved White goat kids.

In sheep, iron deficiency anaemia in indoor-reared lambs has been identified as a risk factor for the development of abomasal bloat and gastropathy [16,17]. Green and Morgan [13] identified anaemia as the primary cause of death in five indoor-reared suckled lambs in a survey on three flocks in south-west England. These deaths occurred between 15 and 30 days of age, and iron deficiency was the suspected cause. 

Vatn and Torsteinbø [18] confirmed a proposed link between iron deficiency in indoor-reared lambs and the occurrence of abomasal bloat. In a preliminary study, they found that lambs which developed abomasal bloat had significantly lower serum iron levels a week before developing the condition than unaffected lambs. A total of 754 twin lambs were included over two years in a trial where one twin was injected subcutaneously with 300 mg iron as iron dextran and the other with sterile saline. Injected lambs showed significantly less abomasal bloat and higher growth rates. As a mechanism the authors suspected that iron deficient lambs develop an abnormal appetite which leads to higher intake of potentially causative bacteria from the soil or bedding. According to Odden et al. [19], 8.3% of sheep farmers in Norway are supplementing iron to newborn lambs with the intention to prevent abomasal bloat, coccidiosis, or both. In an intervention study, the same authors found no difference in the excretion of *Eimeria* spp. oocyst excretion between supplemented and unsupplemented lambs. 

Earlier, Green et al. [20] demonstrated that indoor-reared lambs injected intramuscularly with 300 mg iron as iron dextran showed improved haematological values and significantly higher weaning weights (1 kg). Daily liveweight gains increased by 10 g/day in treated lambs, which was not statistically significant. 

The present study was conducted in two parts: 1.In a preliminary study, a convenience sample of indoor-reared suckled lambs and kids under different management systems were tested for venous blood haemoglobin (Hb) concentration. Comparator samples were taken from lambs and kids fed milk replacer and from outdoor reared suckled lambs. It was hypothesized that lambs and kids reared indoors on maternal milk would show lower haemoglobin levels compared to those reared outdoors or on milk replacer.2.In an intervention study, indoor-reared lambs and kids on whole ewe’s and doe’s milk respectively were randomly assigned to two groups: untreated controls and those receiving iron supplementation by iron dextran injection. Growth rates were compared between test and control animals, as were blood haemoglobin levels pre-treatment and at 1 month of age. It is hypothesised that lambs and kids treated with an iron injection would show higher haemoglobin levels at four weeks of age and higher daily liveweight gains at one and two months of age compared to untreated controls.

## 2. Materials and Methods

### 2.1. Preliminary Study

An early lambing (Dorset x British Milk Sheep x Finn) sheep flock was identified in Oxfordshire, England, where there were large numbers of both artificially reared and suckled lambs, and where lambs were housed for their entire lives. The ewes were fed a grass-silage-based TMR, which was fed out once a day and pushed up multiple times a day. Ewes and lambs were individually housed for the first 24–48 h of life and were group housed thereafter. All pens were straw-bedded. The flock lambed in December and January. This farm also had a smaller group of outdoor lambing ewes of the same breed, that lambed in April while permanently at pasture, so suckled, outdoor-reared, lambs were also available.

A large commercial, mainly Saanen, dairy goat herd was the source of artificially reared, housed kids, while a small, Boer goat herd was the source of suckled, housed kids. Both sets of kids sampled were born in January and February. In both farms, the does received a hay-based ration, supplemented by commercially produced pelleted concentrate feed. On both farms, the goats were housed on straw.

A total of 20 animals from each group were sampled by jugular venepuncture at 1 month of age into EDTA sample tubes. Samples were submitted for haemoglobin analysis to a commercial veterinary laboratory (Axiom Veterinary Laboratories Ltd., Newton Abbot, Devon, UK). The laboratory uses a Sysmex XT2000i-V Manual HAEM007, which is UKAS accredited for sheep.

The results were analysed using the statistics programme Minitab. Hb levels in each group were compared using Student’s *t*-test, and the fraction of animals in each group with Hb levels below the normal range were compared using Fisher’s exact test.

This preliminary study received approval following ethical review by the CRERB of the RVC (URN CR2021-007-2).

### 2.2. Intervention Study

#### 2.2.1. Trial Farms

Three sheep flocks were recruited: the flock from the preliminary study and in addition a pedigree Texel breeder and a dairy sheep flock, mainly using German East Friesian Dairy Sheep. All had in common the fact that lambs were born and reared indoors on whole ewe’s milk. On the first two farms, the lambs were suckled by their mothers until weaning. In the former, the lambs were never turned out; in the latter, ewes and lambs were not turned out until lambs were 8 weeks old. On the sheep dairy, replacement lambs were reared artificially on ad libitum ewe’s milk, with creep feed offered from one week, and they were weaned at about one month and then kept indoors for another two months. 

The Texel flock was fed hay and grass silage and commercially compounded concentrate feed pre-lambing (ration calculated to meet energy and protein requirements based on foetal number pre-lambing) and was housed on straw. After lambing, the diet was ad lib grass silage until turnout, the precise timing of which was affected by weather conditions, but was usually when lambs were around 1.5 months old.

Two goat herds were recruited: the same Boer herd from the preliminary study and a mainly Anglo-Nubian dairy goat herd where doe kids suckled their mothers until weaning at 8 weeks old. The dairy herd was fed a hay-based ration, supplemented with commercially compounded concentrate feed, and was housed on straw. 

#### 2.2.2. Study Protocol

Clinically healthy lambs and kids were recruited between one and eight days of age and randomly (coin toss) assigned to either of two groups, treatment with iron injection (INJ) and the control—no iron injection (CON). On entry to the trial, they were weighed (or the birth weight recorded if available), and the ID (eartag number), sex, date of birth, and whether they were being reared as a twin or a single was recorded. Where twins from the same dam could be identified, the first twin was randomly assigned using the coin, and the second twin assigned to the opposite group. INJ lambs were given 300 mg of iron as iron dextran intramuscularly (1.5 mL Uniferon^TM^, Pharmacosmos, Holbaek, Denmark), and INJ kids were given 150 mg of iron (0.75 mL of the same product) intramuscularly. The intramuscular injections were given in the quadriceps muscle of the hind leg at a single site. The chosen dosages were taken from [21,22] for goats and [18,20] for sheep. The randomisation was carried out on an individual animal level; INJ and CON lambs were kept in the same pens together. Lambs and kids were observed for local or systemic adverse reactions. 

Blood samples were taken from up to 10 control animals and 10 treated animals prior to treatment on each farm. These same animals were resampled after 1 month. Sampling and testing were conducted as in the preliminary study. 

Farms were revisited after 1 month and then after a further month, and all trial animals were re-weighed at these visits.

Haemoglobin baseline levels in the first 8 days of life were compared with haemoglobin levels at one month of age, both the absolute values and the differences between treated and untreated animals. The laboratory reference range value of 8–15 g/dL was used; any animals with Hb levels under 7 g/dL were classed as severely anaemic, treated with iron dextran (if not already done so), and taken out of the study from this point. 

#### 2.2.3. Data Analysis

A sample size calculation for the primary outcome (difference in daily liveweight gain in the first month) was carried out before the study, with the result given in Table 1:

Data were entered into Excel (Microsoft) spreadsheets and analysed in SPSS (IBM SPSS Statistics Version 28.0.0). Data were presented as mean, standard deviation (SD), or the confidence interval (CI), as appropriate. Two-sample independent t-tests were performed to compare 0–1 months, 1–2 months, and 0–2 months DLWG and haemoglobin levels and differences at one month of age. Treatment data and mortality figures were also compared, and tests were chosen where appropriate according to the effect size (chi-squared or Fisher’s exact test).

## 3. Results

### 3.1. Preliminary Study

Mean Hb levels for the artificially reared lambs and kids and the outdoor suckled lambs were within the reference range given by the testing laboratory (8–15 g/dL). Mean Hb levels for the indoor, suckled lambs and kids were below that range (Table 2). There was a difference between the Hb levels of suckled and artificially reared indoor kids (*p* = 0.000) and between indoor, suckled lambs and outdoor and artificially reared lambs (*p* = 0.000, Table 3). A higher fraction of suckled, indoor lambs and kids had Hb levels below the reference range in comparison with the other groups (all *p*-values below 0.01).

### 3.2. Intervention Study

#### 3.2.1. Adverse Effects of the Product

In over 100 lambs and over 30 goat kids treated with iron dextran, 1 lamb showed a transient swelling of the quadriceps muscle (injection site) and mild lameness, but this recovered spontaneously within a few days. It was treated with a single dose of meloxicam (1 mg/kg) administered by subcutaneous injection. No side effects were observed in the other treated animals. 

#### 3.2.2. Lambs

Overall, 234 lambs were originally enrolled between the three farms, 136 from farm 1, 36 from farm 2, and 62 from farm 3. 

##### Daily Liveweight Gain

Mean weights at birth/in the first week of life were 5.57 kg in the INJ group and 5.50 kg in the CON group; the difference was non-significant (*p* = 0.72). 

0–1 month

At one month of age, mean weights were not different at 14.844 kg for INJ and 14.180 kg for CON lambs (*p* = 0.17). 

Average daily liveweight gains (DLWG) were only significantly different for twins and triplets (*p* = 0.047), but not for singles (*p* = 0.463) and overall (*p* = 0.08). The figures per farm and overall are presented in Table 4, Table 5 and Table 6 and Figure 1 and Figure 2.

1–2 months

Between one and two months of age, 211 animals were available for weighing. INJ lambs grew on average 332 g per day, CON 313, a difference of 19 g per day (*p* = 0.222). 

In the twins and triplets (n = 129), INJ animals grew on average 316 g/day, CON animals 291 g/day, a difference of 25 g/day, which was not statistically significant (*p* = 0.185). 

0–2 months

Overall, during the trial period, INJ lambs grew on average by 327 g/day, CON by 310, a difference of 17 g, not statistically significant (*p* = 0.108).

Twins and triplets grew on average 310 g/day in the INJ group and 288 g/day in the CON group, a difference of 22 g per day, not statistically significant (*p* = 0.062). 

##### Haemoglobin Levels

Between the three farms paired blood samples at 0 and 1 month were collected from 54 animals. The remaining six animals either died or were off site, or the samples were clotted and not available for analysis. 

The mean Hb levels in the first week of life were 10.98 g/dL in the INJ group (n = 28) and 10.85 in the CON group (n = 26), showing no difference (*p* = 0.81). Mean Hb levels at one month of age were higher in the INJ group (11.28 g/dL) than in the CON group (10.15 g/dL), *p* = 0.001. 

With a reference range given by the testing laboratory of 8–15 g/dL, 2 out of 26 animals (8%) in the CON group were below the reference range at one month of age, while all of the INJ animals showed Hb levels within the reference range at one month of age (*p* = 0.004).

Haemoglobin levels between the first week and one month raised on average by 0.30 g/dL in the INJ group and fell by 0.7 g/dL in the CON group, with the difference in the change not statistically significant (*p* = 0.11). 

##### Treatments and Mortality

Out of the 234 lambs originally enrolled, 9 died during the trial period—3 from the INJ group and 6 from the CON group. A total of 17 lambs were treated with antibiotics during the trial period—10 from the INJ group and 7 from the CON group. 

#### 3.2.3. Goat Kids

Overall, 67 kids from two farms were enrolled in the trial—31 from farm 1 and 36 from farm 2. 

##### Daily Liveweight Gain

0–1 month

Overall paired weighing data from 56 kids was available.

At first weighing, INJ kids weighed on average 3.738 kg and CON kids 3.937 kg, a difference of 199 g, which was not statistically significant (*p* = 0.270). 

INJ kids grew on average 159 g/day, CON kids 152 g/day, a difference of 7 g per day, which was not statistically significant (*p* = 0.570). The figures per farm are presented in Table 7.

1–2 months

A total of 55 paired weights were available; INJ kids (n = 29) grew on average at 155 g/day, and CON kids (n = 26) by 136 g/day, showing no difference (*p* = 0.179).

0–2 months

During the trial period from 0 to 2 months, INJ kids grew on average by 158 g/day, and CON kids by 144 g/day, showing no difference (*p* = 0.099).

Average weight at 2 months of age

At the end of the trial period, at two months of age, INJ kids weighed on average 13.331 kg and CON kids 12.713 kg, showing no difference (*p* = 0.287).

##### Haemoglobin

At enrolment, INJ kids had mean Hb values of 8.8 g/dL, and CON kids had 9.0 g/dL, showing no difference (*p* = 0.721). 

At one month of age, INJ kids had a mean Hb value of 9.4 g/dL while CON kids had a mean Hb value of 7.6, below the laboratory reference range of 8–15 g/dL, showing a significant difference (*p* = 0.042). 

Hb levels fell on average by 1.5 g/dL between first and second sampling in the CON group, while they rose on average by 0.6 g/dL in the INJ group. However, this difference in change was not significant (*p* = 0.072).

While all Hb values at one month of age in the INJ group were within the reference range (8–15 g/dL), five out of nine untreated goat kids showed values below 8 g/dL.

##### Treatments and Mortality

Of the 67 goat kids originally enrolled, 3 died during the trial period, 1 from the INJ group and 2 from the CON group. Treatment records were available from farm 1, where five kids were treated for coccidiosis—four from the INJ group and 1 from the CON group. Two kids were treated for navel ill—one from the INJ group and one from the CON group. 

## 4. Discussion

### 4.1. Preliminary Study

Haemoglobin concentrations being significantly lower in indoor-reared lambs and kids fed on whole milk is not surprising. Similar findings are well known from calves [6,7,8,9,10] and piglets [5]. Anaemia in indoor-reared suckled lambs has been reported previously. Green et al. [12] reported conjunctival pallor affecting up to 15% of lambs during the first 6 weeks of life in three early lambing, indoor-reared flocks in south-west England. By these authors’ calculations, the sensitivity of conjunctival pallor for anaemia detection was only 53–55%. They determined that iron deficiency was the likely cause, with low serum iron concentrations present. Affected lambs showed anisocytosis, polychromasia, hypochromasia, poikilocytosis, and microcytosis. In a survey, Green and Morgan [13] identified anaemia as the primary cause of death in 0–3.5% of lambs. Similar to this study, they also found a statistically significant difference in haemoglobin levels between indoor-reared and outdoor-reared suckled lambs. In piglets, this is widely attributed to rooting and ingestion of soil iron, although there is increasing evidence that outdoor pigs benefit from iron supplementation [23]. Whether a similar pattern applies to lambs is unclear, and the data have only been collected on a single farm. 

This study appears to be the first to compare haemoglobin levels in indoor-reared lambs on maternal milk to those receiving milk replacer. The results confirm that the higher iron levels in milk replacer result in less anaemia in the lambs receiving it compared to whole maternal milk. The same was found in goats by this study; however, there is the additional complicating factor that the suckled and milk-replacer fed kids were of different breeds and on different holdings, so the difference in diet cannot be conclusively shown to be the cause of the difference in anaemia levels.

Overall, it can be concluded that similar to calves and piglets, an indoor environment with no access to soil iron combined with the low iron content of maternal milk can lead to subclinical anaemia.

### 4.2. Intervention Study

#### 4.2.1. Lambs

Overall daily liveweight gains were consistently higher in treated lambs on all sample farms, but neither on individual farms nor overall was this difference statistically significant. On the sample size calculation, a difference of 20 g/day and a standard deviation of 50 g/day was assumed; in the present study, the actual difference in the means was 18 g/day, and the standard deviation was 79 g/day in the INJ group and 76 g/day in the CON group. On the basis of these results, a sample size of 340 per group would have been required in order to detect a difference at the 0.05 significance level with 80% probability. In comparison, Vatn and Torsteinbø [18] worked with a larger sample size and found a significant difference in growth rates from birth to slaughter in treated (300 mg iron) and untreated lambs. Green et al. [20] used about 190 lambs in each group and found a significant difference in growth rates in the pre-weaning period (220 g vs. 200 g). Those figures and the dosage were used for the sample size calculation and the dosage regime in the present study. The main reasons for the increased standard deviation in the current study are likely to be farm factors (e.g., farm 2 fed whole ewe’s milk artificially ad libitum and achieved considerably higher daily liveweight gains than the other two farms) and the fact that singles and twins/triplets were analysed together, while the two quoted studies were carried out on pairs of twins only. When individual farms were analysed separately, the standard deviations decreased, but so did the sample size, and the differences, while numerically consistent in favour of the INJ group, were again not statistically significant. However, for twins overall, a significant difference in DLWG could be established, but the reason for this remains unclear. It may be that the reason for the twins showing a larger effect than singles is that singles tend to have heavier birthweights than twins, and so received a lower mg/kg dose of iron dextran. In pigs, heavier piglets have been shown to be more likely to have markers of anaemia at weaning than lighter piglets in farms supplementing iron through a standard dose of iron dextran in the first few days of life [24,25].

The haemoglobin levels in untreated animals at one month of age were generally higher than in the preliminary study, but the levels in treated and untreated animals were comparable to the levels established by Green et al. [20] at 24 days in the UK and lower than those obtained by Vatn and Torsteinbø [18] in Norway, but in both trials the difference between treated and untreated animals was significant.

Mortality and treatment data were only presented descriptively due to the small numbers. The incidence of abomasal bloat, a condition linked to iron deficiency by [18], was very low on all participating farms.

#### 4.2.2. Goat Kids

The small available sample size (below the suggested numbers from the sample size calculation) and large variation in first weights made a meaningful analysis of daily liveweight gains difficult. However, as in the lambs, injected kids showed significantly higher Hb values than control animals at one month of age. Whether this is clinically relevant should be subject of further work involving more animals.

## 5. Conclusions

Iron dextran injection at 300 mg iron in neonatal lambs led to higher haemoglobin levels at one month of age and a significant increase in daily liveweight gain in twins and triplets. In goat kids given 150 mg of iron, haemoglobin levels at one month of age were significantly higher than in untreated controls, with daily liveweight gains numerically, but not significantly raised in treated animals. This study indicates that anaemia in indoor-reared lambs and kids on maternal milk is widespread and can be corrected with iron dextran at the given dosages, leading to a measurable increase in daily liveweight gain in twin and triplet lambs. 

## Figures and Tables

**Figure 1 animals-12-01863-f001:**
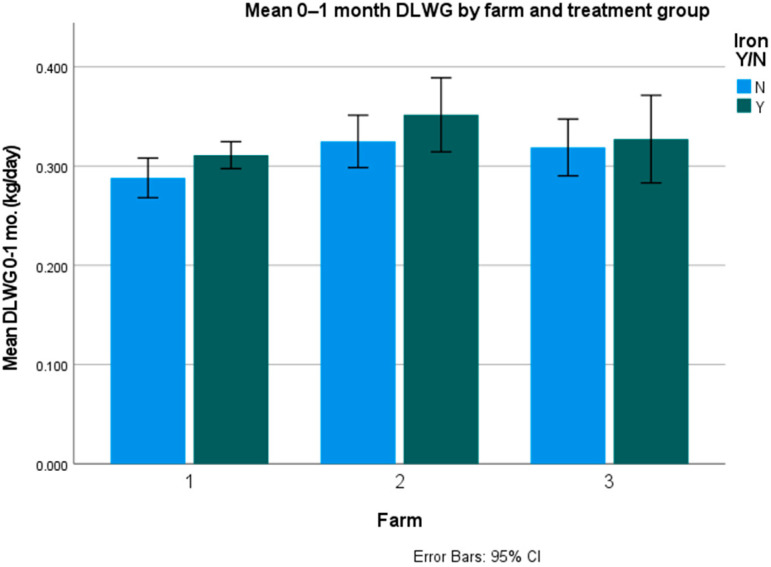
Mean overall DLWG on the farms with error bars (kg/day).

**Figure 2 animals-12-01863-f002:**
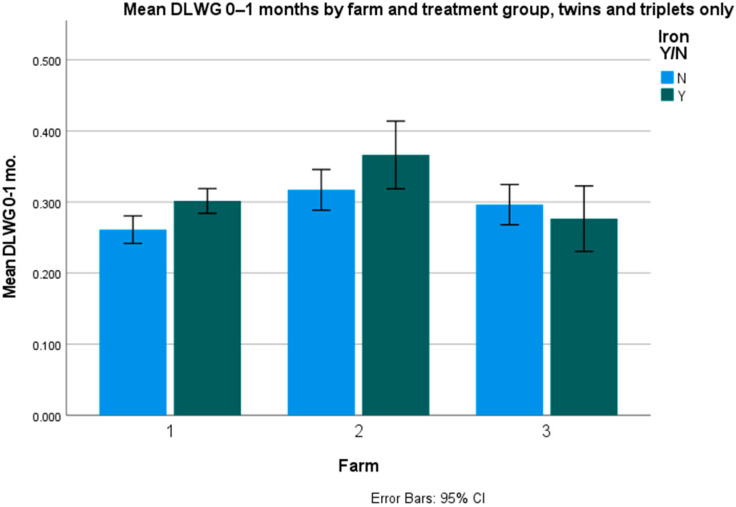
Mean 0–1-month DLWG in twins and triplets (kg/day).

**Table 1 animals-12-01863-t001:** Sample size calculation, based on a *p*-value of 0.05 and a power of 0.8; calculator used: https://www.sealedenvelope.com/power/continuous-superiority/ (accessed on 18 November 2021).

	Kids	Lambs
Assumed average DLWG treated (g/d)	160	220
Assumed average DLWG untreated (g/d)	145	200
Assumed standard deviation (g/d)	28	50
Sample size per group required	55	99
Total sample size required	110	198

**Table 2 animals-12-01863-t002:** Mean haemoglobin concentrations of the tested groups of lambs and kids from the preliminary study. The 95% confidence interval is also shown, as is the percentage of animals in each group with a haemoglobin concentration below the reference range of 8–14 g/dL.

Group	Species	Environment	Diet	Mean [Hb] (g/dL)	95% CI	% [Hb] < Ref. Range
1	Goat	Indoors	Milk replacer	9.65	9.07–10.23	5%
2	Goat	Indoors	Suckling	7.73	7.21–8.24	45%
3	Sheep	Indoors	Milk replacer	10.13	9.37–10.89	5%
4	Sheep	Indoors	Suckling	6.8	6.14–7.46	80%
5	Sheep	Outdoors	Suckling	9.08	8.49–9.66	20%

**Table 3 animals-12-01863-t003:** Results of the comparison of the concentration of haemoglobin by Student’s *t*-test, and the comparison of the percentage of animals with haemoglobin concentration below the reference range by Fisher’s exact test between the different groups. Cross-species comparisons were not performed.

Species	Comparison	Variable	*p*-Value
Goats	Indoor: suckled vs. milk replacer	[Hb]	0.000
Sheep	Indoor: suckled vs. milk replacer	[Hb]	0.000
Sheep	Suckled: indoor vs. outdoor	[Hb]	0.000
Sheep	Outdoor suckled vs. indoor milk replacer	[Hb]	0.037
Goats	Indoor: suckled vs. milk replacer	% [Hb] < ref.range	0.008
Sheep	Indoor: suckled vs. milk replacer	% [Hb] < ref.range	0.000
Sheep	Suckled: indoor vs. outdoor	% [Hb] < ref.range	0.000
Sheep	Outdoor suckled vs. indoor milk replacer	% [Hb] < ref.range	0.342

**Table 4 animals-12-01863-t004:** Overall DLWG in lambs between 0–1 month.

Farm		0–1 Month DLWG (g/day)		Difference	*p*-Value
	n	INJ (SD)	CON (SD)		
1	125	311 (54)	288 (78)	23	0.058
2	31	352 (65)	325 (51)	27	0.208
3	61	327 (118)	319 (78)	8	0.743
**Total**	**217**	**321 (79)**	**303 (76)**	**18**	**0.08**

**Table 5 animals-12-01863-t005:** Overall DLWG in lambs, 0–1 month, singles only.

Farm		0–1 Month DLWG (g/Day)		Difference	*p*-Value
	n	INJ (SD)	CON (SD)		
1	49	328 (50)	324 (89	4	0.868
2	7	315 (48)	361 (53)	−46	0.287
3	27	393 (121)	346 (94)	47	0.265
**Total**	**83**	**348 (84)**	**334 (88)**	**14**	**0.463**

**Table 6 animals-12-01863-t006:** Mean 0–1-month DLWG in twins and triplets on the three farms.

Farm		0–1 Month DLWG (g/Day)		Difference	*p*-Value
	n	INJ (SD)	CON (SD)		
1	76	301 (55)	261 (56)	40	0.002
2	24	366 (67)	317 (50)	49	0.05
3	34	277 (90)	296 (55)	−20	0.444
**Total**	**134**	**305 (71)**	**282 (59)**	**23**	**0.047**

**Table 7 animals-12-01863-t007:** Overall DLWG in kids between 0 and 1 month.

Farm	n	0–1 Month DLWG (g/Day)		Difference	*p*-Value
		INJ (SD)	CON (SD)		
1	22	200 (33)	191 (32)	9	0.508
2	34	129 (28)	129 (33)	0	0.956
**Total**	**56**	**159 (46)**	**152 (44)**	**7**	**0.570**

## Data Availability

All raw data are available from the authors.

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
