# Peer review of "Anaemia in Lambs and Kids Reared Indoors on Maternal Milk and the Impact of Iron Supplementation on Haemoglobin Levels and Growth Rates"

_animals, 2022, doi:10.3390/ani12141863_

Round 1
Reviewer 1 Report
This study assessed anaemia in newborn lambs and goat kids by hemoglobin levels and examined the effects of iron dextran injection on daily gain in lambs and goat kids. The topic of this paper has certain significance for improving breeding management measures. In-depth knowledge of the etiology of lamb anemia and implement effective prevention measures is essential to improve lamb health. However, the contents of this paper is not so novel, and the measurement index is too simple. In addition, the experiment design is not clear enough, and there are many problems in the presentation of results and data.
Line 49 to Line 66: In the introduction, the authors proposed a potential relationship between iron deficiency and abomasal bloat, but there were no data related to gsatropathy in the results.
Materials and methods: The experimental design involves species, environment and diet, but in materials and methods, the description of the experimental design and the grouping of experimental animals is not clear enough. In addition, there is a lack of description of replicates in intervention trial.
Line 122: In the treatment groups, lambs were given 300 mg iron, but goat kids were given 150 mg iron. What was the basis for this treatment?
Line 168: It is recommended to merge Table 1 and Table 2.
Table 3 and Figure 1 present the same data, and only one of them can be retained. Table 4 and Figure 2 have the same problem.
From line 249 to line 333: In the results section, many of the descriptions have no data source, neither figures nor tables. All data should be presented in figures or tables.
Reviewer 2 Report
The paper submitted for editorial check describes in a clear way how the administration of an iron dextran had a positive influence on the level of hemoglobin in the blood in one month old lambs and goat kids. The research was conducted in 2 stages and covered a preliminary study from which the results were used to select a study group of lambs and goat kids for the intervention study.
I would suggest that the authors present the research stages in a diagram which would help readers understand the research steps.
Referring to the preliminary study some questions arise :- What breed of lambs were used (meat producing or milk producing)? In what conditions were the „indoor” group held (were they in individual enclosures with their mothers or in a herd with other mothers)? What kind of feed were the mothers given and how many times per day were they fed? Were the outdoor group kept at pasture the entire day? Can the authors please supply the geographical location from where the animals were obtained? During which season of the year was the study conducted? Was the study done in the grazing season or outside of the grazing season.
With regards to the „intervention study” part of the article please clarify if the iron dextran was administered in a single dose? Were animals that were displaying side effects after dextran injection removed from the study? If yes, please supply this information in the paper. I am also interested if the authors, in this study or any other studies, observed if the meat or milk quality was affected by the administration of iron dextran?
Editorial remarks : In the entire article please use either g/day or grams/day. Please remove unnecessary “rounded” brackets () for example on line 42, 46, 339 and initials W.O. on line 54.
Reviewer 3 Report
Page 2, line 79: The introduction paragraph ends abruptly. Additionally, there is no objective or hypothesis. A hypothesis must be present for the reader to understand the purpose, direction, and thoughts of the authors to set the stage for the remainder of the manuscript. In its current state, this paper has no direction or scientific question. This must be present prior to publication.
Page 3, line 82: The material and methods section is extremely brief. A lot of detail has been eliminated from this section that would be considered beneficial in order for others to replicate this study (i.e., description of the indoor housing facilities, how and what were all animals fed/offered in their environments which may have influenced haemoglobin value, maternal diet, bedding substate, etc.). Please provide more detail on animal management/housing/nutrition so this experiment could be conducted again.
Page 3, line 104: Was lamb or kid breed a part of your statistical model? How could breed or crossbred vs. purebred impact results shown?
Page 3, line 105: Describe whole milk. I assume that this is sheep milk and not from any other species.
Page 3, line 122: You state the supplemental iron was given intramuscularly. Please elaborate where this was performed on the body (i.e., area or muscle) as well as if there were any adverse effects of the injection (i.e., adverse reactions, visual pain or distress at injection area, negative effects on meat quality). Edit – I see that this detail is available on Page 7, lines 198-202. Please consider moving to the section listed within this comment for clarity.
Page 4, lines 128-130: This section is repeated/copied from the preliminary section. Consider removing or making it briefer to avoid repetition.
Page 4, line 139: There is no mention of what your level of significance (p-value) is. In most cases, anything > 0.05 is considered significant; however, unless stated, this assumption can not be used. Additionally, please describe your model in detail. How were animals blocked and how were
Page 4, lines 142-143: A table should be able to stand alone without any other reference. Please provide a title and definitions for abbreviations used within the table. Additionally, the 5th line in the table is blank making it look odd. Please add a header or remove to maintain consistency.
Page 5, line 164: Avoid using the phrase ‘statistically significant’. When describing data, you should only be reporting on differences that are significantly different. Rather, please add a p-value at the end of the statement to demonstrate significance. Furthermore, please add a p-value at the end of each statement in results sections to demonstrate significance between selected comparisons. Edit – this is done correctly beginning on Page 7. Please follow the same format as done here for consistency and remove the terms ‘statistically significant’ as it is redundant.
Page 5, line 168: Table headings/descriptions should be placed above the table whereas figure descriptions are placed below. Please revise.
Page 6, line 190: Table 2 appears messy because of colors and wordy text. Consider changing to white and black and abbreviate in the table. Define abbreviations below table.
Page 7, line 209: Odd wording and unnecessary detail. Please remove or reword.
Page 7, lines 211-213: Consider rewording for clarity and to remove the use of ‘statistically significant’. (i.e., At one month of age mean lamb BW for INJ and CON lambs were not different (14.8 kg vs. 14.2 kg)).
Page 7, lines 206-232: Very choppy and messy. Is reporting a specific difference each time important to the overall objective/goal of the research? Please clean this section up using the recommendations previously provided. Please revise entire results section – most was skipped because of inconsistency and difficulty in reading/interpretation.
Page 11, line 342: Are the authors referencing the FAMACHA eye scoring system here? Regardless, please explain the use of this technique and how it relates to your work.
Page 12, lines 352-358: I would like some discussion and explanation upon why the authors believe that there is a difference between sheep raised with their dams housed indoors compared with lambs artificially reared indoors or with their dams outdoors. This seems to be a major oversight in this piece.
Page 13, line 399: There seems to be very little discussion/explanation through out the manuscript about goat kids and makes it appear as it was an afterthought. Can this portion be removed to focus of sheep – which appears to be the main focus already and then submit the goat information separately or as supplemental data? In the current format, the goat information seems out of place.
Page 13, lines 405-411: The conclusion is poorly written and does not provide any new interpretation or understanding of the data set or information that has been previously provided elsewhere in the manuscript. A conclusion provides high points with interpretation of this data. I would like to see this in addition to how this information can be applied on-farm for the benefit of small ruminants and their producers.
Round 2
Reviewer 1 Report
The experiments performed in this manuscript to my opinion remain too preliminar and descriptive. The major weakness of this manuscript is the writing, there are numerous problems. The fixed effect in this experiment included species, environment and diet, but statistical method does not fully address the multivariate nature of the data. The presentation of the experimental data still needs to be improved. Many of the data mentioned in the results (1-2 month, 0-2 month, etc.) are not presented in either figure or table. The current structure of the paper makes it difficult to read. As such, in my opinion, the manuscript is still not suitable for publication.